# Phenome-wide Mendelian randomization study of plasma triglyceride levels and 2600 disease traits

Joshua K Park[1,2,3†], Shantanu Bafna[1,2†], Iain S Forrest[1,2,3], Áine Duffy[1,2], Carla Marquez-Luna[1,2], Ben O Petrazzini[1,2], Ha My Vy[1,2], Daniel M Jordan[1,2], Marie Verbanck[4], Jagat Narula[5,6], Robert S Rosenson[5,7], Ghislain Rocheleau[1,2], Ron Do[1,2]*

[1]Charles Bronfman Institute for Personalized Medicine, Icahn School of Medicine at Mount Sinai, New York, United States; [2]Department of Genetics and Genomic Sciences, Icahn School of Medicine at Mount Sinai, New York, United States; [3]Medical Scientist Training Program, Icahn School of Medicine at Mount Sinai, New York, United States; [4]Université Paris Cité, Paris, France; [5]Department of Medicine, Icahn School of Medicine at Mount Sinai, New York, United States; [6]Cardiovascular Imaging Program, Zena and Michael A. Wiener Cardiovascular Institute, Mount Sinai Heart, Icahn School of Medicine at Mount Sinai, New York, United States; [7]Metabolism and Lipids Unit, Zena and Michael A. Wiener Cardiovascular Institute, Mount Sinai Heart, Icahn School of Medicine at Mount Sinai, New York, United States

*For correspondence:
ron.do@mssm.edu

†These authors contributed equally to this work

## Abstract

**Background:** Causality between plasma triglyceride (TG) levels and atherosclerotic cardiovascular disease (ASCVD) risk remains controversial despite more than four decades of study and two recent landmark trials, STRENGTH, and REDUCE-IT. Further unclear is the association between TG levels and non-atherosclerotic diseases across organ systems.

**Methods:** Here, we conducted a phenome-wide, two-sample Mendelian randomization (MR) analysis using inverse-variance weighted (IVW) regression to systematically infer the causal effects of plasma TG levels on 2600 disease traits in the European ancestry population of UK Biobank. For replication, we externally tested 221 nominally significant associations (p<0.05) in an independent cohort from FinnGen. To account for potential horizontal pleiotropy and the influence of invalid instrumental variables, we performed sensitivity analyses using MR-Egger regression, weighted median estimator, and MR-PRESSO. Finally, we used multivariable MR (MVMR) controlling for correlated lipid fractions to distinguish the independent effect of plasma TG levels.

**Results:** Our results identified seven disease traits reaching Bonferroni-corrected significance in both the discovery (p<1.92 × 10$^{-5}$) and replication analyses (p<2.26 × 10$^{-4}$), suggesting a causal relationship between plasma TG levels and ASCVDs, including coronary artery disease (OR 1.33, 95% CI 1.24–1.43, p=2.47 × 10$^{-13}$). We also identified 12 disease traits that were Bonferroni-significant in the discovery or replication analysis and at least nominally significant in the other analysis (p<0.05), identifying plasma TG levels as a novel potential risk factor for nine non-ASCVD diseases, including uterine leiomyoma (OR 1.19, 95% CI 1.10–1.29, p=1.17 × 10$^{-5}$).

**Conclusions:** Taking a phenome-wide, two-sample MR approach, we identified causal associations between plasma TG levels and 19 disease traits across organ systems. Our findings suggest unrealized drug repurposing opportunities or adverse effects related to approved and emerging TG-lowering agents, as well as mechanistic insights for future studies.

**Funding:** RD is supported by the National Institute of General Medical Sciences of the National Institutes of Health (NIH) (R35-GM124836) and the National Heart, Lung, and Blood Institute of the NIH (R01-HL139865 and R01-HL155915).

## Editor's evaluation

In this work causal associations between plasma triglyceride (TG) levels and 19 disease traits were identified which may provide valuable insights into the mechanism of TG biology and drug repurposing of TG-lowering agents. The evidence supporting the claims of the authors is solid, based on biobank-scale data in both discovery analysis and replication analysis. The work will be of interest to cardiovascular clinicians, medical geneticists, and pharmaceutical companies.

## Introduction

ASCVD remains the leading cause of death worldwide despite the effectiveness of statin therapy in reducing low-density lipoprotein cholesterol (LDL-C) levels (*Baigent et al., 2005*; *Roth et al., 2020*). Additional therapeutic targets and adjunctive treatments are needed to address the burden arising from residual risk (*Rosenson and Goonewardena, 2021*). TGs play vital roles in physiology, ranging from energy storage and mobilization to inflammation, thrombosis, and hormone-like signaling (*Zewinger et al., 2020*; *Norata et al., 2007*). However, a causal relationship between TGs and ASCVDs remains controversial (*Miller et al., 2011*; *Albrink and Man, 1959*), recently culminating in the conflicting reports of two double-blinded randomized controlled trials (RCT), STRENGTH, and REDUCE-IT (*Nicholls et al., 2020*; *Bhatt et al., 2019*; *Doi et al., 2021*). Nevertheless, drug development for reducing TG-rich lipoproteins (TRL) is an active area of research and several targets have now been validated, including angiopoietin-like 3 (ANGPTL3) (*Graham et al., 2017*; *Dewey et al., 2017*; *Musunuru et al., 2010*; *Gaudet et al., 2017*), angiopoietin-like 4 (ANGPTL4) (*Dewey et al., 2016*), and apolipoprotein C-III (APOC3) (*Gaudet et al., 2015*; *Crosby et al., 2014*). Clinical trials are currently evaluating these targets for dyslipidemias (*Arrowhead, 2021b*; *Arrowhead, 2021a*).

Whether TGs are causal risk factors or simply associative biomarkers remains uncertain not only for ASCVDs but also for other diseases of different organ systems. Understanding the causal effects of TGs across a broader range of human diseases could have significant implications for drug repurposing. TG-lowering agents, such as fibrates (*Frick et al., 1987*; *Ginsberg et al., 2010*) and omega-3 fatty acids *Bhatt et al., 2019* have already been approved; however, not knowing which diseases are causally affected by TGs precludes their use for indications other than hypertriglyceridemia. Understanding the protective effects of TGs could also have implications for drug safety. With the recent approval of icosapent ethyl (*Bhatt et al., 2019*; *Gaba et al., 2022*) and the ongoing development of other TG-lowering agents (*Shaik and Rosenson, 2021*), such as ANGPTL3 (*Graham et al., 2017*; *Dewey et al., 2017*), ANGPTL4, and APOC3 inhibitors (*Gaudet et al., 2015*), long-term safety becomes a matter of concern. Post-market surveillance data are limited; therefore, identifying diseases with negative causal links to TGs could suggest adverse side effects, whereas protective causal links to TGs could suggest therapeutic avenues and indices informative for drug development. Furthermore, this could inform polypharmacy and drug titration in clinical practice.

Several methodological challenges have prohibited causal conclusions about TGs across human diseases. First, TGs are correlated with established ASCVD risk factors, such as obesity and insulin resistance (*Eckel et al., 2005*). They also correlate with LDL particles or apolipoprotein B (apoB) concentration (*Cromwell et al., 2007*) and inversely correlated with high-density lipoprotein cholesterol (HDL-C) (*Phillips and Smith, 1991*). Conventional observational studies have thus been limited in drawing causal inferences due to potential confounding. Second, available TG-lowering agents have pleiotropic effects on several major lipid fractions, including VLDL-C, LDL-C, HDL-C, and apolipoproteins, including APOC3 (*Ginsberg et al., 2010*; *Keech et al., 2005*; *Frick et al., 1987*). Thus, costly RCTs have had limited power to disentangle the effects of lowering TG specifically (*Sarwar et al., 2010*; *Goldberg et al., 2011*; *Bhatt et al., 2019*). Finally, MR studies have typically focused on individual diseases selected a priori (*Davey Smith and Hemani, 2014*), narrowing the scope of causal estimates to ASCVDs while overlooking non-ASCVD diseases (*Harrison et al., 2018*; *Smith et al., 2014*; *Allara et al., 2019*).

Phenome-wide MR is a high-throughput extension of MR that estimates the causal effects of an exposure on multiple outcomes simultaneously. As in conventional MR, this method uses genetic variants as instrumental variables (IV) to proxy modifiable exposures (*Smith and Ebrahim, 2003*), and importantly, it relies on three critical assumptions: (1) The genetic variant is directly associated with the exposure; (2) The genetic variant is unrelated to confounders between the exposure and outcome; and (3) The genetic variant has no effect on the outcome other than through the exposure (*Smith and Ebrahim, 2003*). A distinction, however, is that phenome-wide MR enables comprehensive scans of the phenotypic spectrum, limiting bias from prior assumptions and facilitating the discovery of unforeseen causal relationships. This has recently become feasible with the maturation of large-scale biobanks providing extensive genetic and phenotypic data, such as the UK Biobank (UKB) (*Bycroft et al., 2018*) and FinnGen project (*FinnGen, 2020*).

Here, we use phenome-wide MR to systematically estimate the causal effects of plasma TGs on 2600 disease outcomes in UKB, followed by replication testing in FinnGen. We then apply multiple MR methods for sensitivity analyses and MVMR to control for plasma LDL-C, HDL-C, and ApoB levels.

## Methods
### Study design and data sources
We followed the guidelines published by *Burgess et al., 2020* and the Strengthening the Reporting of Observational Studies in Epidemiology (STROBE) guidelines (*Skrivankova et al., 2021*). Accordingly,

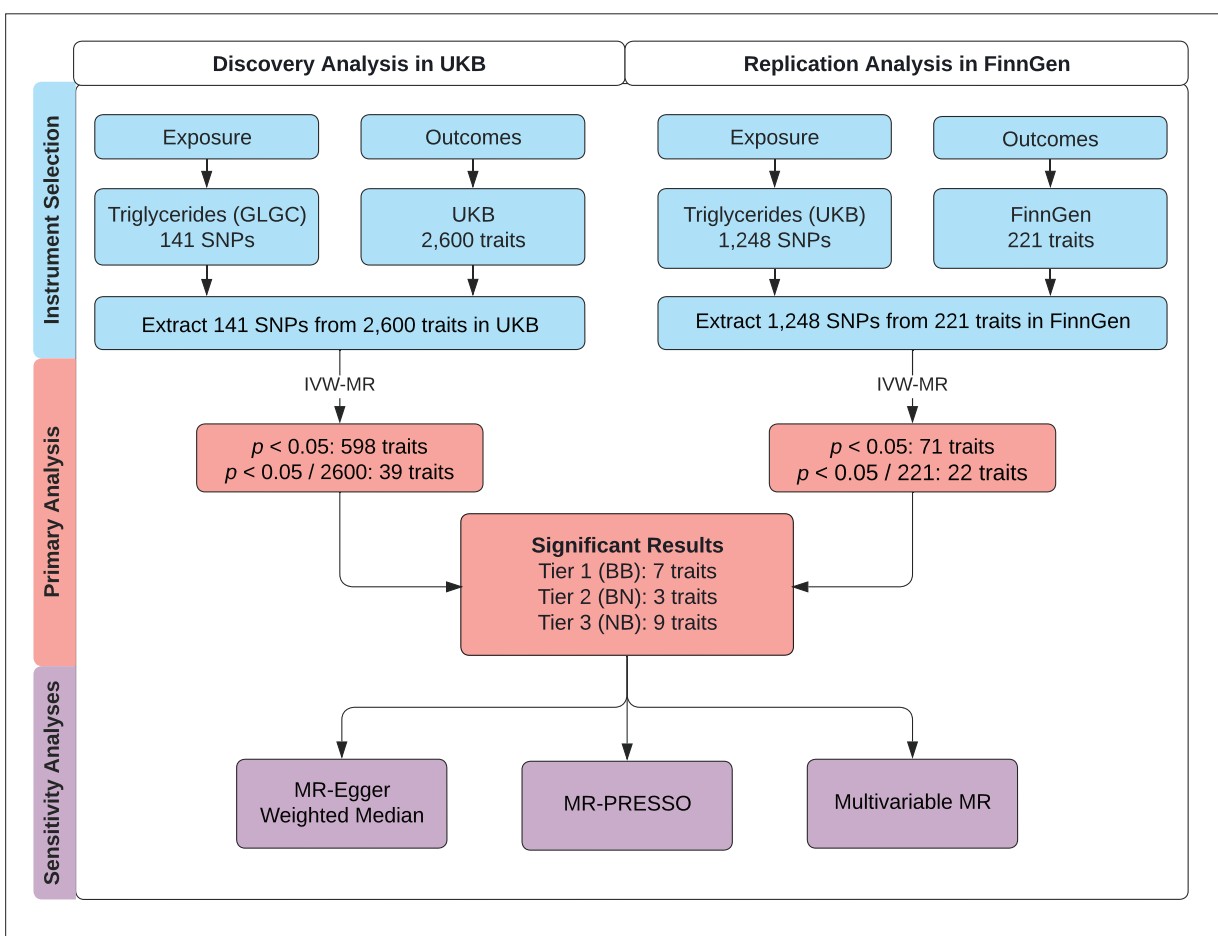

**Figure 1.** Study overview. A schematic summarizing the study design. GLGC, Global Lipids Genetics Consortium; HDL, high-density lipoprotein; IVW, inverse-variance weighted; LDL, low-density lipoprotein; MR, Mendelian randomization; MR-PRESSO, Mendelian Randomization Pleiotropy RESidual Sum and Outlier; SNP, single-nucleotide polymorphism; TG, triglyceride; UKB, UK Biobank. Tier 1 (BB): At least Bonferroni-significant in both the discovery and replication analyses. Tier 2 (BN): At least Bonferroni-significant in the discovery analysis and at least nominally significant in the replication analysis. Tier 3 (NB): At least nominally significant in the discovery analysis and at least Bonferroni-significant in the replication analysis.

we note that MR analyses rely on three important instrumental variable assumptions: (1) the genetic variant is directly associated with the exposure; (2) the genetic variant is unrelated to confounders between the exposure and outcome; and (3) the genetic variant has no effect on the outcome other than through the exposure (*Smith and Ebrahim, 2003*). We consider these assumptions more thoroughly in the discussion. This study uses three non-overlapping genome-wide association studies (GWAS). A schematic figure summarizes the study design (*Figure 1*).

For primary and sensitivity analyses, we conducted two-sample MR taking summary statistics for genetic associations with plasma TGs from one dataset (Global Lipids Genetics Consortium, GLGC) (*Willer et al., 2013*) and summary statistics for genetic associations with outcomes from a second, independent dataset (UKB). The CARDIoGRAMplusC4D Metabochip study by GLGC involved 63,746 cases and 130,681 controls (*Deloukas et al., 2013*). UKB is a longitudinal, population-based cohort study with genetic and phenotypic data on over 500,000 participants aged 40–69 years at recruitment from across the United Kingdom during 2006–2010 (*Sudlow et al., 2015*). The sociodemographic and health-related characteristics of UKB participants have been described elsewhere (*Fry et al., 2017*).

For replication analysis, we again performed two-sample MR but used summary statistics for TG from UKB and summary statistics for outcomes from FinnGen (Release 4; 176,899 samples; 169,962,023 variants). FinnGen is a large public-private partnership started in August 2017 aiming to collect and analyze genomic and phenotypic data from 500,000 Finnish biobank participants (*FinnGen, 2020*; *Kurki et al., 2022*). For both primary and replication analysis, there is likely to be minimal sample overlap in the two-sample MR, which can cause weak-instrument bias and inflated type 1 error (*Burgess et al., 2016*).

## Genetic instruments for plasma TG levels

For primary and secondary analyses, we identified 3086 single nucleotide polymorphisms (SNP) associated with plasma TG levels in GLGC at a genome-wide significance threshold of $p<5 \times 10^{-8}$. A total of 141 independent SNPs were then selected at a linkage disequilibrium (LD) threshold of $r^2 <0.05$ using the 1000 Genomes LD European panel as the reference population (*Auton et al., 2015*). To select independent SNPs, we used the PLINK software's LD clumping command (*Purcell et al., 2007*) with the following options: `--clump-p1 0.00000005 --clump-p2 0.0005 --clump-r2 0.05 --clump-kb 250`. We note that the GLGC GWAS had excluded individuals known to be on lipid-lowering medications (*Willer et al., 2013*).

For replication analysis, we identified 1388 SNPs associated with plasma TG levels in UKB after restricting for genome-wide significance ($p<5 \times 10^{-8}$) and LD-clumping. The SNPs were selected at an LD threshold of $r^2 <0.05$ using the 1000 Genomes LD European panel as the reference population. LD clumping was performed by PLINK using the same command as above. We then restricted to 1248 SNPs for genetic instrumentation based on whether the SNP was also available in the FinnGen dataset. We note that the Pan-UKB GWAS study did not exclude participants based on their use of lipid-lowering medications.

## Genetic instruments for disease outcomes

As a phenome-wide investigation, this study examined many binary disease outcomes. For outcomes of primary and sensitivity analyses, we used genome-wide association summary statistics from the European ancestry subset of UKB. The focus on European ancestry was needed to reduce heterogeneity and maximize statistical power. Pan-UKB had performed 16,131 GWASs on 7221 phenotypes in ~420,531 UKB participants of European ancestry using genetic and phenotypic data (*PanUKBTeam, 2020*). A total of 7221 total phenotypes had been categorized as 'biomarker,' 'continuous,' 'categorical,' 'ICD-10 code,' 'phecode,' or 'prescription' (*PanUKBTeam, 2020*). We filtered for outcomes to retain categorical, ICD-10, and phecode types; non-null heritability in European ancestry as estimated by Pan-UKB; and relevance to disease, excluding medications. This yielded 2600 traits for the primary analysis. The exact sample size of each GWAS for each of these traits is provided in *Supplementary file 1*.

For outcomes of replication analysis, we used genome-wide association summary statistics from FinnGen. To allow for compatibility between UKB and FinnGen datasets, phenotypes coded as FinnGen 'endpoint IDs' were mapped to ICD-10 codes or phecodes corresponding to the coding

convention used by UKB. Categorical traits in UKB could not be reliably mapped to FinnGen IDs, requiring us to omit categorical traits from replication analysis. Of 2444 available FinnGen outcomes, we ultimately selected 221 outcomes for replication testing based on two conditions: (1) The outcome has an equivalent phenotype documented as an ICD-10 code or phecode in UKB, which we had included in the discovery analysis, and (2) the outcome was at least nominally significant (p<0.05) in the discovery analysis.

## Statistical analyses

Primary discovery analysis used a fixed-effect inverse-variance weighted (IVW) method of two-sample MR over the more horizontal pleiotropy-robust MR-Egger regression method to maximize statistical power (*Burgess et al., 2013*). To proxy TG, we selected the 141 TG-associated SNPs from GLGC identified above as instrumental variables in MR testing. All SNPs were harmonized to match the effect and non-effect allele at each SNP. For replication analyses, we conducted IVW tests using TG-associated SNPs from UKB as exposures and 221 traits at least nominally significant (p<0.05) in the discovery analysis as outcomes from FinnGen. We selected UKB TG GWAS loci as the instruments for replication on FinnGen outcomes, rather than GLGC TG GWAS loci, to diversify the source of TG instruments and mitigate potential biases associated with one TG GWAS. Moreover, UKB GWAS included a larger study population than GLGC GWAS, providing a greater number of genetic instruments that can together explain more of the variance in plasma TG levels, and thus, greater statistical power and precision. Nevertheless, we also performed the replication analyses using TG instruments from GLGC and included these results as supplemental data (*Supplementary file 5*). R (4.1.0) was used for all analyses (*R Development Core Team, 2021*).

For sensitivity analyses, we addressed the possible presence of horizontal pleiotropy by applying MR-Egger (*Bowden et al., 2015*), weighted median estimator (*Bowden et al., 2016*), and MR-PRESSO outlier tests (*Verbanck et al., 2018*) on Bonferroni-significant traits identified in the primary analysis above. The utility of these various methods has been extensively reviewed (*Davies et al., 2018*). MR-PRESSO outlier tests were performed on 16 traits with significant MR-PRESSO global test results (p<0.05). To account for potential horizontal pleiotropy due to outliers, we examined the MR-PRESSO outlier test p-values from the discovery analysis, and as recommended (*Burgess et al., 2020*), removed outlier IVs (between 1 and 10 SNPs for each outcome) based on a corrected significance threshold (p<0.05 / 141=3.55 × 10$^{-4}$). IVW tests were then re-run without the outlier IVs in the discovery analysis.

Additionally, we fitted MVMR models controlling for plasma LDL-C, HDL-C, and ApoB levels in the discovery stage. MVMR uses genetic variants associated with multiple exposures to estimate the effect of each exposure on an outcome, thereby accounting for potentially correlated exposures (*Sanderson, 2021*). We extracted the 141 LD-independent TG-associated SNPs described above from the summary statistics for LDL-C and HDL-C from GLGC and for ApoB from UKB. We then performed four MVMR IVW tests against traits Bonferroni-significant in our discovery analysis with the following sets of exposures: TG and LDL; TG and HDL; TG, HDL, and LDL; or TG, HDL, LDL, and ApoB.

Finally, we performed bidirectionally or reverse MR on significant results to examine the potential presence of reverse causation. We selected instruments for each disease as described above from Pan-UKB and instruments for plasma TG levels from GLGC, essentially reversing the discovery stage design using a fixed-effect IVW method.

To account for multiple testing, a conservative Bonferroni threshold for statistical significance was set to p<0.05/2600=1.92 × 10$^{-5}$ for the discovery analysis. For replication analysis, a similar conservative Bonferroni threshold was set to p<0.05/221=2.26 × 10$^{-4}$. We determined three tiers of statistical evidence based on the Bonferroni or nominal threshold (p<0.05) in both the discovery and replication analyses. This presentation is somewhat unconventional and partly arises from the study's use of three different datasets for instrument selection. In a traditional two-stage discovery and replication design, Bonferroni adjustment is based on the number of significant signals from the discovery that is tested in replication. Here, we used three tiers of statistical evidence to present results because a standard meta-analysis between UKB and FinnGen was not possible, given it was not possible to reliably map all phenotypes between the two datasets. Additionally, Bonferroni-significant results in the replication analysis would have been ignored in FinnGen in a sequential two-stage design if they were also only nominally associated in UKB. The three tiers are defined below:

1. Tier 1 (BB): At least Bonferroni-significant in both the discovery and replication analyses.

2. Tier 2 (BN): At least Bonferroni-significant in the discovery analysis and at least nominally significant in the replication analysis.
3. Tier 3 (NB): At least nominally significant in the discovery analysis and at least Bonferroni significant in the replication analysis.

## Ethical approval

UK Biobank has approval from the North West Multi-Centre Research Ethics Committee (MREC) as a Research Tissue Bank (RTB) (11/NW/0382), and all participants of UKB provided written informed consent. More information is available at (https://www.ukbiobank.ac.uk/learn-more-about-uk-biobank/about-us/ethics). The work described in this study was approved by UKB under application number 16218. All participants of FinnGen provided written informed consent for biobank research, based on the Finnish Biobank Act. The Coordinating Ethics Committee of the Hospital District of Helsinki and Uusimaa (HUS) approved the FinnGen study protocol Nr HUS/990/2017. More information is available at (https://www.finngen.fi/en/code_of_conduct).

## Results

### Genetically proxied plasma TG levels and phenome-wide disease risk

In the discovery analysis, we identified nominally significant associations (p<0.05) between plasma TG levels and 598 disease traits in UKB (*Supplementary file 1*). Of these, 39 disease traits were statistically significant after multiple testing corrections with a conservative Bonferroni-corrected threshold (p<1.92 × 10^{-5}). As a positive control, plasma TG levels were positively associated with gout with an odds ratio (OR) of 1.78 for gout (95% CI 1.52–2.09, p=7.41 × 10^{-11}), in agreement with prior studies (*Yu et al., 2021*). In the replication analysis, we identified nominally significant associations (p<0.05) between plasma TG levels and 71 disease traits in FinnGen (*Supplementary file 2*). Of these, 22 traits were Bonferroni-significant. A summary of the 19 most statistically significant and replicated results has been organized into three predefined tiers of evidence (*Figure 2*). We note that the magnitude of estimates from the discovery analysis was generally greater than those from replication analysis, especially among tier 1 and tier 2 associations, potentially as a manifestation of winner's curse bias (*Göring et al., 2001*).

For tier 1 results, genetically determined plasma TG levels were positively associated with seven disease traits in both the discovery and replication analyses. These were Bonferroni-significant in both analyses, and all were related to dyslipidemias or ASCVD. Among traits related to ASCVD, the strongest association by statistical significance was for angina pectoris in both the discovery UKB cohort (OR 1.39, 95% CI 1.29–1.51, p=2.11 × 10^{-13}) and the replication FinnGen cohort (OR 1.30, 95% CI 1.24–1.37, p=1.25 × 10^{-26}). For tier 2 results, plasma TG levels were positively associated with three disease traits, including non-ASCVDs: gout, uterine leiomyoma, and 'other aneurysms' (phecode-442). The strongest association by significance, after gout, was for leiomyoma of the uterus in both the discovery (OR 1.19, 95% CI 1.10–1.29, p=1.17 × 10^{-5}) and replication cohorts (OR 1.06, 95% CI 1.02–1.11, p=3.60 × 10^{-3}). For tier 3 results, nine disease traits were identified, and a greater proportion were non-ASCVD traits. The strongest association by significance was for hypertension in both the discovery (OR 1.51, 95% CI 1.08–1.23, p=3.20 × 10^{-5}) and replication cohorts (OR 1.17, 95% CI 1.14–1.22, p=9.49 × 10^{-20}).

### Sensitivity analyses using MR-PRESSO, MR-Egger, weighted median, and reverse MR

To account for potential horizontal pleiotropy bias due to outlier IVs, we next used the MR-PRESSO test on the 19 significant and replicated associations identified above, which were categorized into three tiers of statistical evidence. MR-PRESSO suspected outlier IVs for 16 of the 19 associations, and between 1 and 10 IVs were identified for eight associations (*Supplementary file 3*). We then re-ran IVW MR for these eight associations after removing outlier IVs. Among tier 1 results, all seven associations increased in significance after outlier removal (*Figure 3*). For tier 2 results, 2 of 3 associations had significant global MR-PRESSO test values, but no outlier IVs were removed. For tier 3 results, seven of nine associations had significant global MR-PRESSO test values, and outlier IVs were removed for one of these associations, which increased in significance: hypertension (p=3.20 × 10^{-5} to p=2.80 × 10^{-8}).

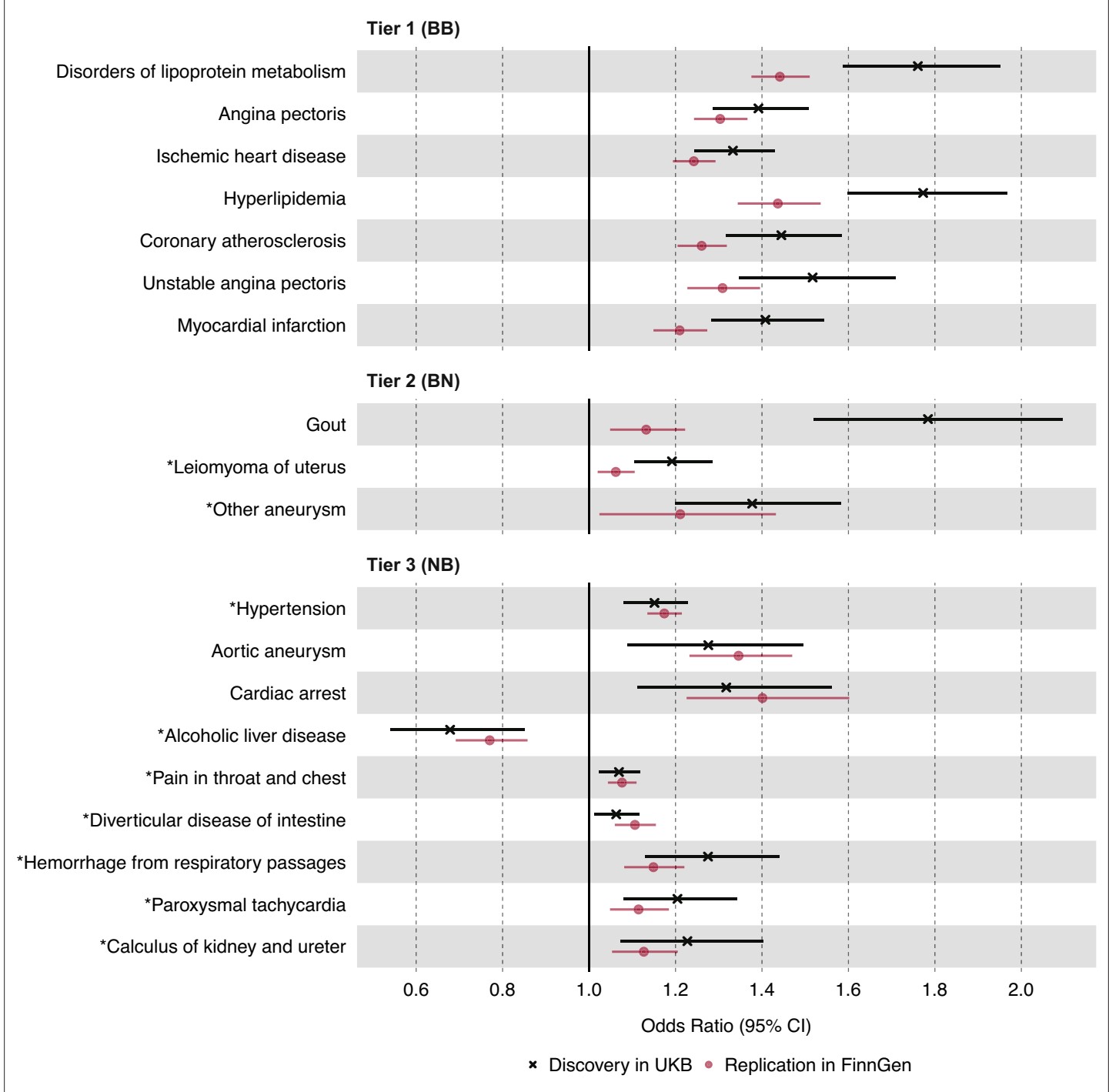

**Figure 2.** Causal estimates of genetically proxied plasma triglyceride (TG) levels on disease risk using inverse-variance weighted (IVW) regression in UKB and FinnGen. Causal estimates from IVW regression are shown as odds ratios (OR) per 1 SD increase in plasma TG levels (mmol/L). Asterisks indicate novel associations. Black indicates discovery analysis results using UKB. Red indicates replication analysis results using FinnGen. Horizontal error bars represent 95% CIs. Tier 1 (BB): At least Bonferroni-significant in both the discovery (p<1.92 × 10⁻⁵) and replication (p<2.26 × 10⁻⁴) analyses. Tier 2 (BN): At least Bonferroni-significant in the discovery analysis (p<1.92 × 10⁻⁵) and at least nominally significant in the replication analysis (p<0.05). Tier 3 (NB): At least nominally significant in the discovery analysis (p<0.05) and at least Bonferroni significant in the replication analysis (p<2.26 × 10⁻⁴). Full results and sample sizes (n) of each GWAS for each trait are provided in *Supplementary file 1* and *Supplementary file 2*.

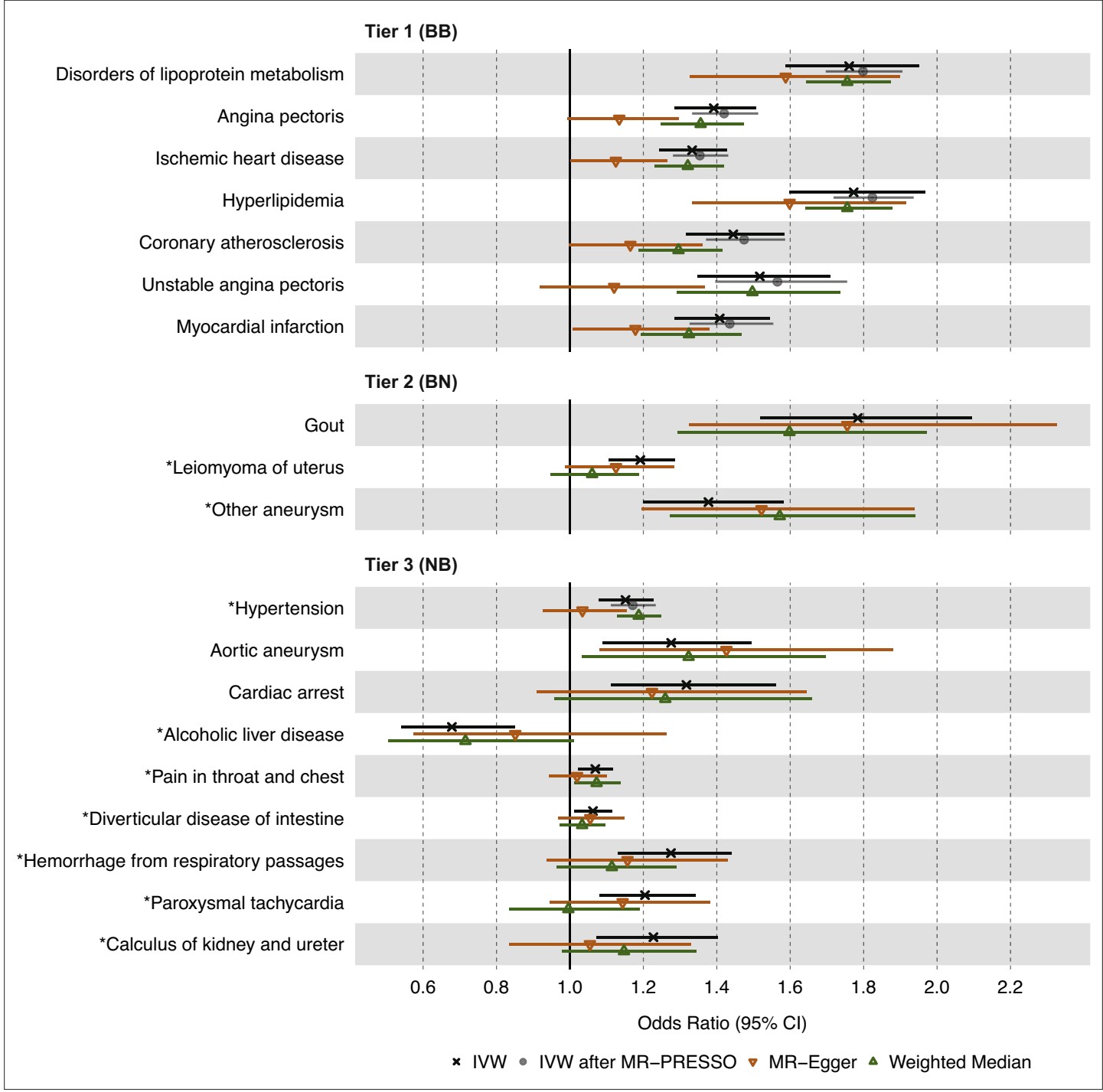

**Figure 3.** Causal estimates of genetically proxied plasma triglyceride (TG) levels on disease risk using MR-Egger, Weighted Median, and Mendelian randomization pleiotropy residual sum and outlier (MR-PRESSO) methods in UKB. Shown are sensitivity analysis results from the discovery stage using instruments from Global Lipids Genetics Consortium (GLGC) and outcomes from UKB. Associations with insignificant global MR-PRESSO test results (p>0.05) were not rerun and do not have data points for inverse-variance weighted (IVW) after MR-PRESSO. Tier 1 (BB): At least Bonferroni-significant in both the discovery (p<1.92 × $10^{-5}$) and replication (p<2.26 × $10^{-4}$) analyses. Tier 2 (BN): At least Bonferroni-significant in the discovery analysis (p<1.92 × $10^{-5}$) and at least nominally significant in the replication analysis (p<0.05). Tier 3 (NB): At least nominally significant in the discovery analysis (p<0.05) and at least Bonferroni significant in the replication analysis (p<2.26 × $10^{-4}$). Horizontal error bars represent 95% CIs. Full results and sample sizes (n) of each GWAS for each trait are found in *Supplementary file 1* and *Supplementary file 3*.

Importantly, all associations maintained the same effect direction after the MR-PRESSO outlier test, in keeping with the initial discovery analysis.

To further account for horizontal pleiotropy, we conducted sensitivity analyses for the 19 significant and replicated traits using the MR-Egger and weighted median estimators on UKB data (*Figure 3*). MR-Egger results were comparable to IVW results for all 19 traits across the three tiers of evidence. However, weighted median results were less statistically significant than IVW and MR-Egger results for most traits except for disorders of lipid metabolism, hyperlipidemia, and hypertension.

Finally, we performed reverse MR to estimate the effects of significant disease traits on plasma TG levels, selecting instruments from UKB and GLGC, respectively. Genetic data were sufficiently available to perform this analysis for 9 of the 19 diseases of interest. These results are presented in *Supplementary file 6*. Expectedly, 'disorders of lipoprotein metabolism' and 'hyperlipidemia' had positive effects on plasma TG levels; however, no other examined disease trait showed results suggesting reverse causation.

## Separating the independent effects of plasma TG levels

To isolate the independent effect of plasma TG levels from those of correlated lipid fractions, we next conducted MVMR on the 19 significant and replicated associations using the IVW estimator on UKB data, adjusting for plasma HDL-C, LDL-C, or both simultaneously (*Figure 4*). For tier 1 associations, we observed an increase in statistical significance for disorders of lipid metabolism and hyperlipidemia after adjustment. For tier 2 results, we observed an increase in significance for gout. For tier 3 results, we observed an increase in significance for alcoholic liver disease, paroxysmal tachycardia, and calculus of the kidney and ureter. Aside from these, MVMR led to a decrease in statistical significance for the remaining associations. However, all 19 associations remained in the same direction of causality despite multivariable adjustment (*Supplementary file 4*).

## Discussion

We performed phenome-wide, two-sample MR to estimate the causal effects of plasma TG levels on a spectrum of human disease traits (n=2600) using a discovery cohort from UKB and a replication cohort from FinnGen. We report seven disease traits reaching Bonferroni-corrected significance in both the discovery and replication analyses, which we categorized as tier 1 results. These traits were predominantly manifestations of ASCVD, such as ischemic heart disease, angina pectoris, and myocardial infarction. Using MVMR, we found that these associations remained even after controlling for LDL-C, HDL-C, and ApoB levels (*Figure 4*). We also identified three disease traits that are Bonferroni-significant in discovery (p<1.92 × 10⁻⁵) and at least nominally significant in replication (p<0.05), categorized as tier 2 results. Lastly, we identified nine disease traits at least nominally significant in discovery and Bonferroni-significant in replication, categorized as tier 3 results. Several of these non-ASCVD disease traits have never been reported to be causally associated with TGs, potentially introducing new opportunities for drug repurposing and experimental study.

Our results are consistent with prior work suggesting that plasma TG levels are causally associated with ASCVD risk (*Sarwar et al., 2010*; *Castañer et al., 2020*; *Do et al., 2013*; *Dewey et al., 2016*; *Holmes et al., 2015*; *White et al., 2016*; *Varbo et al., 2013*; *Ibi et al., 2021*; *Rosenson et al., 2021*). We do not prove here that circulating TGs per se are directly atherogenic; rather, we show that plasma TG measurement, which comprises multiple classes of triglyceride-rich lipoproteins (TRL) and their remnants, captures a clinically significant mechanism that is causally associated with ASCVD (*Rosenson et al., 2021*). This interpretation is consistent with an emerging view that apolipoprotein B (ApoB) particle number is a more important determinant of atherogenesis than LDL-C and that TRL particles are as important a risk factor in ASCVD as LDL particles since they each carry a single ApoB molecule (*Ference et al., 2019*; *Marston et al., 2022*; *Richardson et al., 2020*). According to this paradigm, the causal association we observe between plasma TG levels and ASCVD risk may be mediated by the concentration of ApoB particles in TRLs, captured by plasma TG level measurements. We evaluated this possibility by accounting for ApoB levels in a MVMR framework and found that significant causal associations persist between plasma TG levels and all examined ASCVD traits, despite a modest decrease in effect size (*Figure 4*; *Supplementary file 4*). These data suggest that ApoB levels may explain at least a portion of the detected associations; however, plasma TG measurement

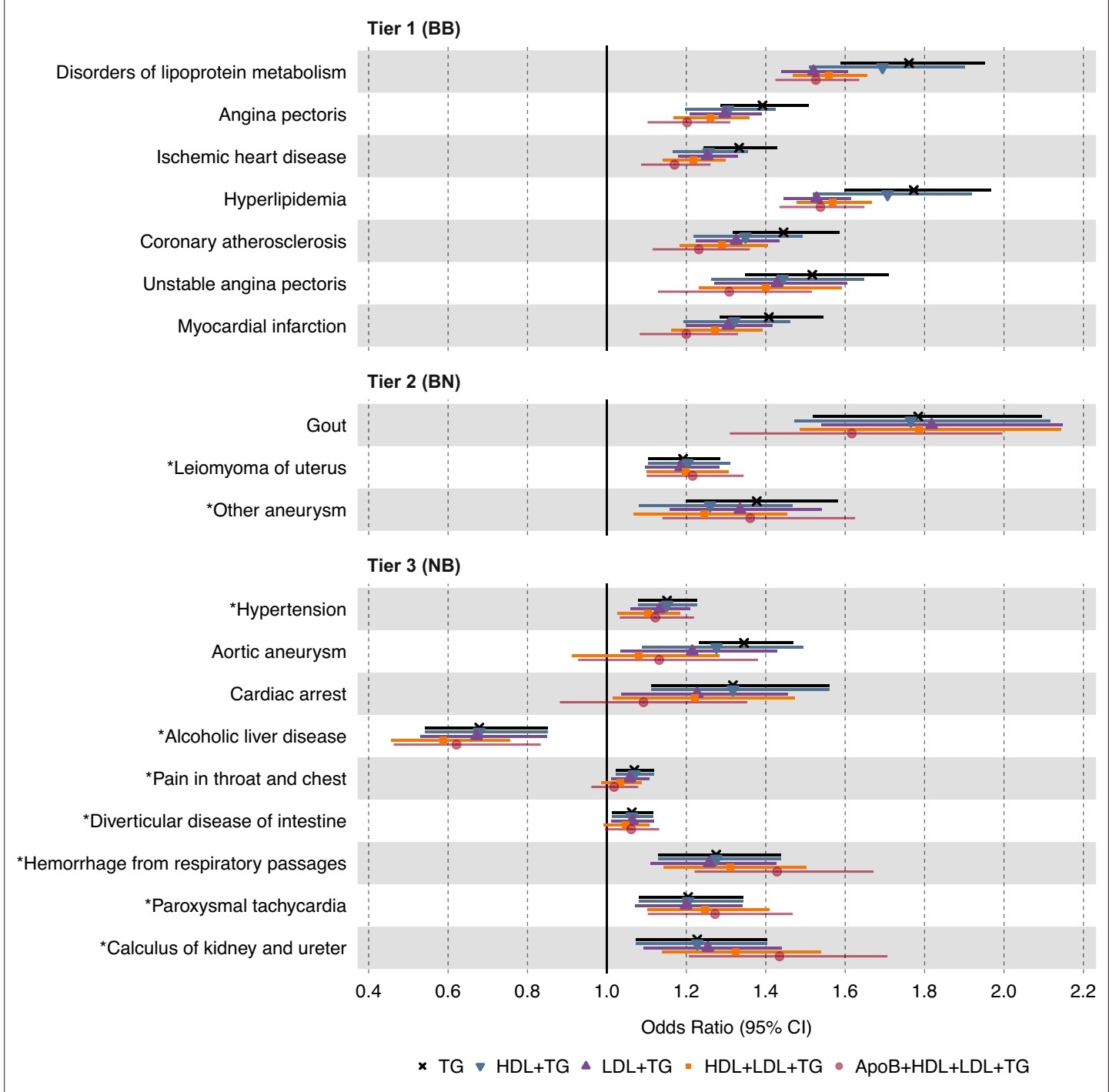

**Figure 4.** Causal estimates of genetically proxied plasma triglyceride (TG) levels on disease risk using multivariable inverse-variance weighted (IVW) regression controlling for plasma low-density lipoprotein cholesterol (LDL-C), high-density lipoprotein cholesterol (HDL-C), and apolipoprotein B (ApoB) levels in UKB. Shown are the multivariable MR (MVMR) results from the discovery analysis, using genetic instruments for plasma TG levels from Global Lipids Genetics Consortium (GLGC) and disease traits from UKB. Levels of statistical significance are categorized into three tiers: Tier 1 (BB): At least Bonferroni-significant in both the discovery ($p<1.92 \times 10^{-5}$) and replication ($p<2.26 \times 10^{-4}$) analyses. Tier 2 (BN): At least Bonferroni-significant in the discovery analysis ($p<1.92 \times 10^{-5}$) and at least nominally significant in the replication analysis ($p<0.05$). Tier 3 (NB): At least nominally significant in the discovery analysis ($p<0.05$) and at least Bonferroni significant in the replication analysis ($p<2.26 \times 10^{-4}$). Horizontal error bars represent 95% CIs. Full results are provided in **Supplementary file 4**. Sample sizes (n) of each GWAS for each trait are found in **Supplementary file 1**.

appears to capture additional mechanisms of ASCVD risk involving cholesterol-enriched TRL remnants that remain unclear. Scientific interest in the atherogenic mechanisms of TRLs continues to grow, as recently reviewed elsewhere (*Borén et al., 2022*; *Ginsberg et al., 2021*).

Regarding non-ASCVDs, we present suggestive genetic evidence of potentially causal associations between plasma TG levels and uterine leiomyomas (uterine fibroids), diverticular disease of the intestine, paroxysmal tachycardia, hemorrhage from respiratory passages (hemoptysis), and calculus of kidney and ureter (kidney stones). Due to the weaker statistical evidence supporting these associations, special caution is encouraged when interpreting these results to infer causality, and further replication and validation studies are essential for all tier 2 and 3 results. Nevertheless, prior studies have reported correlational associations between many of these diseases and plasma TG levels or risk factors correlated to plasma TGs. For example, studies had documented positive correlations between leiomyoma risk and plasma TG levels (*Uimari et al., 2016*; *Tonoyan et al., 2021*; *Peshkova et al., 2020*). Others had shown diverticular disease risk is positively associated with BMI, body fat percentage, and visceral fat area (*Shih et al., 2022*; *Freckelton et al., 2018*; *Böhm, 2021*). Similarly, kidney stones have been associated with the triglyceride-glucose (TyG) index (*Qin et al., 2021*), and atrial tachycardias have been associated with VLDL levels in metabolic syndrome patients (*Lee et al., 2017*; *Park and Lee, 2018*). Our study is the first to suggest that these associations may be causally related and specific to plasma TG levels, as opposed to confounding risk factors, such as obesity and hormone replacement therapy. However, we note that these associations were not as robust as those in tier 1 associations related to ASCVDs, and we encourage caution in drawing causal conclusions from these estimates of a causal effect. Though, we present the rationale for potentially repurposing TG-lowering agents towards these non-ASCVD indications and for pursuing mechanistic studies interrogating TG biology, RCTs remain the gold standard for assessing causality and remain necessary for assessing clinical potential.

Our results also suggest a novel, negative causal association between plasma TG levels and alcoholic liver disease (ALD), suggesting that excessively reducing plasma TG levels may increase the risk of this disease trait. A mechanistic explanation for this association remains elusive as previous studies on dysregulated lipid metabolism in liver disease have generally focused on non-alcoholic fatty liver disease (NAFLD). However, one animal study suggests that medium-chain TGs may decrease lipid peroxidation and reverse established alcoholic liver injury in rat models of ALD (*Nanji et al., 1996*). Nevertheless, that elevated TG levels may protect against ALD is surprising and requires external validation. It remains unclear whether this association is only relevant to lifelong, chronic lowering of plasma TG levels rather than transient lowering by drug-based interventions. It is also unclear whether ALD could be prevented by increasing TG levels. This finding suggests the potential intolerability of TG-lowering agents and the importance of maintaining these drugs' concentrations within therapeutic windows for patient safety; however, we reiterate that this tier 3 association was only nominally significant in discovery, while Bonferroni-significant in replication, and future studies are needed to validate the statistical evidence.

The study has several limitations. First, MR is a powerful but potentially fallible method that relies on several key assumptions, namely that genetic instruments are (i) associated with the exposure (the relevance assumption); (ii) have no common cause with the outcome (the independence assumption); and (iii) have effects on the outcome solely through the exposure (the exclusion restriction assumption) (*Hartwig et al., 2016*). In MR, (i) is relatively straightforward to test, while (ii) and (iii) are difficult to establish unequivocally. As a prominent example, horizontal or type I pleiotropy has been shown to be common in genetic variation, which can bias MR estimates (*Verbanck et al., 2018*; *Jordan et al., 2019*). This occurs when a genetic instrument is associated with multiple traits other than the outcome of interest. To detect and correct for this as best as possible, we used various MR tests as sensitivity analyses that each aims to adjust for or account for the presence of horizontal pleiotropy, including MR-PRESSO, as well as MR-Egger and weighted median methods. There is no universally accepted method that is perfectly robust to horizontal pleiotropy, but we take the best current approach by using multiple methods and examining the consistency of results. A second limitation of the study is that genetic variants confer exposures that are lifelong but small in effect size; thus, MR may over- or underestimate the effect sizes of pharmacological interventions. MR also cannot make comparisons between TG-lowering agents as it evaluates drug targets, not drug subclasses and unique pharmacodynamics (*Gill et al., 2019*; *Sofat et al., 2010*). However, the utility of MR in drug target validation is

well-established as estimates still indicate the presence and direction of causality (*Gill et al., 2021*). Third, UKB and FinnGen have innate differences in participant demographics and medical coding systems, due in part to the former being based in the United Kingdom and the latter in Finland. As such, the potential misclassification of participants in the case-control assignment is a liability to this study. We exercised caution in mapping UKB traits to FinnGen traits, but we were unable to reliably map *all* 'categorical' traits from UKB to corresponding traits in FinnGen, testing for replication only 221 of the 598 associations that were nominally significant in the primary analysis. We note however that, despite geographical differences, both datasets largely involve White European participants of older age, with the mean age in UKB and FinnGen being 56.5 and 59.8, respectively. Fourth, discovery and replication analyses were not completely independent, since UKB data were used in both analyses. This could potentially exacerbate demographic and measurement biases inherent to UKB; however, we show that taking a traditional replication approach using GLGC instead of UKB for selecting exposure instruments in replication returns comparable tier 1 results (*Supplementary file 5*), while losing statistical power to highlight many of the tier 2 and 3 results. Fifth, the GLGC GWAS used to select instruments for plasma TG levels in discovery had accounted for lipid-lowering treatment, while the UKB GWAS used in replication had not. Last, we acknowledge that this study was restricted to populations of European ancestry, limiting the generalizability of our findings. A recent study observed comparable MR estimates for the causal effects of lipid traits on ischemic stroke risk between African and European ancestry individuals (*Fatumo et al., 2021*), suggesting the potential generalizability of MR results across ancestries in some specific cases. Nevertheless, trans-ancestry MR analyses are warranted to validate our study's findings in diverse ancestry populations.

In conclusion, this study demonstrates a high-throughput application of two-sample MR to estimate the causal effects of plasma TG levels on phenome-wide disease risk. Future studies may consider the functional and qualitative attributes of TRL subtypes, and metabolomic data partitioning TRLs by size and composition may soon enable this to identify the specific component of a plasma TG measurement that drives disease risk in the detected associations (*Holmes et al., 2015*). With the proliferation of multi-omic data, this systematic MR approach could be generalized to study the causal effects of serum biomarkers on disease risk, at scale. However, caution is still warranted in inferring causality, as MR depends on specific assumptions and the validity of those assumptions must be carefully assessed. Thus, diverse study designs remain necessary to triangulate evidence on the causal effects of plasma TG levels.

## Acknowledgements

We acknowledge the participants and investigators of the GLGC, FinnGen, and UKB studies.

## Additional information

### Competing interests

Robert S Rosenson: reports receiving grants from Amgen, Arrowhead, Lilly, Novartis and Regeneron; consulting fees from Amgen, Arrowhead, Lilly, Novartis and Regeneron; honoraria for non-promotional lectures from Amgen, Kowa and Regeneron, royalties from Wolters Kluwer (UpToDate); and stock holdings in MediMergent, LLC. Ron Do: reports receiving grants from AstraZeneca; grants and non-financial support from Goldfinch Bio; being a scientific co-founder, consultant, and equity holder (pending) for Pensieve Health; and a consultant for Variant Bio, all unrelated to this work. The other authors declare that no competing interests exist.

### Funding

| Funder | Grant reference number | Author |
| --- | --- | --- |
| National Institute of General Medical Sciences | R35-GM124836 | Ron Do |
| National Heart, Lung, and Blood Institute | R01-HL139865 | Ron Do |

| Funder | Grant reference number | Author |
| --- | --- | --- |
| National Heart, Lung, and Blood Institute | R01-HL155915 | Ron Do |

The funders had no role in study design, data collection and interpretation, or the decision to submit the work for publication.

## Author contributions
Joshua K Park, Shantanu Bafna, Data curation, Formal analysis, Investigation, Writing - original draft, Writing – review and editing; Iain S Forrest, Formal analysis, Validation, Investigation, Writing – review and editing; Áine Duffy, Resources, Validation, Methodology, Writing – review and editing; Carla Marquez-Luna, Supervision, Validation, Investigation, Writing – review and editing; Ben O Petrazzini, Resources, Supervision, Investigation, Methodology, Writing – review and editing; Ha My Vy, Resources, Data curation, Supervision, Methodology, Writing – review and editing; Daniel M Jordan, Supervision, Investigation, Methodology, Writing – review and editing; Marie Verbanck, Investigation, Methodology, Writing – review and editing; Jagat Narula, Resources, Data curation, Supervision, Writing – review and editing; Robert S Rosenson, Supervision, Validation, Methodology, Writing – review and editing; Ghislain Rocheleau, Resources, Data curation, Supervision, Investigation, Methodology, Writing – review and editing; Ron Do, Conceptualization, Data curation, Formal analysis, Supervision, Funding acquisition, Investigation, Methodology, Project administration, Writing – review and editing

## Author ORCIDs
Joshua K Park http://orcid.org/0000-0002-1719-3537
Ghislain Rocheleau http://orcid.org/0000-0002-9989-7553
Ron Do http://orcid.org/0000-0002-3144-3627

## Ethics
UK Biobank has approval from the North West Multi Centre Research Ethics Committee (MREC) as a Research Tissue Bank (RTB) (11/NW/0382), and all participants of UKB provided written informed consent. More information is available at (https://www.ukbiobank.ac.uk/learn-more-about-uk-biobank/about-us/ethics). The work described in this study was approved by UKB under application number 16218. All participants of FinnGen provided written informed consent for biobank research, based on the Finnish Biobank Act. The Coordinating Ethics Committee of the Hospital District of Helsinki and Uusimaa (HUS) approved the FinnGen study protocol Nr HUS/990/2017. More information is available at (https://www.finngen.fi/en/code_of_conduct).

## Decision letter and Author response
Decision letter https://doi.org/10.7554/eLife.80560.sa1
Author response https://doi.org/10.7554/eLife.80560.sa2

# Additional files

## Supplementary files
• Supplementary file 1. Causal estimates of plasma TG levels on 2600 traits in UKB using multiple MR methods. Shown are the estimates, standard deviations, and $p$-values of MR results using IVW, MR-Egger, and weighted median methods. 141 SNPs were used as instrumental variables to proxy plasma TG levels. A Bonferroni threshold for statistical significance was set to $p < 0.05/2600 = 1.92 \times 10^{-5}$ for this discovery analysis.

• Supplementary file 2. Causal estimates of plasma TG levels on 221 traits in FinnGen using multiple MR methods. Shown are the estimates, standard deviations, and $p$-values of MR results using IVW, MR-Egger, and weighted median methods. 1248 SNPs were used as instrumental variables to proxy plasma TG levels. A Bonferroni threshold for statistical significance was set to $p < 0.05/221 = 2.26 \times 10^{-4}$ for this replication analysis.

• Supplementary file 3. IVW-MR estimates of significant and replicated associations (tier 1–3) after MR-PRESSO outlier tests using UKB data. Shown are the estimates, standard deviations, and $p$-values of IVW-MR results in the discovery analysis, before and after outlier IV removal, for significant and replicated traits (tier 1–3) that had significant MR-PRESSO global test results ($p < 0.05$) in the primary IVW analysis of the discovery UKB cohort.

• Supplementary file 4. Multivariable IVW-MR estimates of plasma TG levels on significant and replicated associations (tier 1–3) using UKB data. Shown are the estimates, standard deviations, and *p*-values of multivariable IVW-MR results controlling for HDL-C, LDL-C, ApoB levels, or all three, in the discovery stage. Only tier 1–3 results significant and replicated as predefined in the methods were examined in this analysis.

• Supplementary file 5. IVW-MR estimates of significant associations (tier 1–3) using GLGC data for exposure instruments and FinnGen data for outcome instruments. Shown are the estimates, standard deviations, and *p*-values of IVW-MR results when using GLGC data to select instruments for plasma TG levels, as was done in the discovery analysis.

• Supplementary file 6. Bidirectional MR estimates of significant associations (tier 1–3) using multiple MR methods. Shown are the estimates, standard deviations, *p*-values, MR-PRESSO outlier test results, and the number of instruments used in bidirectional MR analyses estimating the causal effects of nine disease traits on plasma TG levels using IVW, MR-Egger, and Weighted Median methods. Exposure instruments were selected from UKB, and outcome instruments were selected from GLGC.

• MDAR checklist

• Reporting standard 1. STROBE checklist.

## Data availability

All data generated in this study are included in the manuscript and supplementary tables. All analyses used publicly available data (UKB , FinnGen), including previously published GWAS (GLGC) (Willer et al., 2013). Obtaining access to UKB (Pan-UKB_Team, 2020) and FinnGen (FinnGen, 2020) GWAS summary statistics is detailed here (https://www.finngen.fi/en/access_results) and here (https://pan.ukbb.broadinstitute.org/downloads). Please note the summary statistics for FinnGen and Pan-UKB are made publicly available.

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
