## [Editor Report]

In this work causal associations between plasma triglyceride (TG) levels and 19 disease traits were identified which may provide valuable insights into the mechanism of TG biology and drug repurposing of TG-lowering agents. The evidence supporting the claims of the authors is solid, based on biobank-scale data in both discovery analysis and replication analysis. The work will be of interest to cardiovascular clinicians, medical geneticists, and pharmaceutical companies.

---

## [Decision Letter]

**Decision letter after peer review:**

Thank you for submitting your article "Phenome-wide Mendelian randomization study of plasma triglycerides and 2,600 disease traits" for consideration by *eLife*. Your article has been reviewed by 3 peer reviewers, one of whom is a member of our Board of Reviewing Editors, and the evaluation has been overseen by YM Dennis Lo as the Senior Editor. The following individual involved in review of your submission has agreed to reveal their identity: Kaixiong Ye (Reviewer #3).

Essential revisions:

1) Additional analysis may be needed to rule out the presence of reverse causation and potential false positive findings.

2) The assumptions of Mendelian randomization need to be discussed in more detail.

3) It is a strength to have a discovery stage and a replication stage. However, the difference between the two stages is blurred by the use of three tiers of statistical evidence. This may increase false positive rates. Further multiple testing adjustment may be needed.

4) The interpretation of the causal effects may need to be more cautious and rigorous, particularly for the findings from Tiers 2 and 3.

*Reviewer #3 (Recommendations for the authors):*

I have another concern or suggestion. One possible interpretation of the causal roles of triglycerides, the authors proposed, "the causal association we observe between plasma TG levels and ASCVD risk may be mediated by the concentration of apoB particles in TRLs, captured by plasma TG level measurements." The possible mediating role of apoB could be tested in the same multivariable MR framework used to exclude the effect of LDL-C and HDL-C. Since there are existing GWAS for apoB, it should be relatively easy to implement this analysis, either using the GWAS of apoB in UK Biobank and then outcome GWAS in FinGenn, or other GWAS of apoB (not in UKB) and then outcome GWAS in UKB.

---

## [Author Response]

Essential revisions:1) Additional analysis may be needed to rule out the presence of reverse causation and potential false positive findings.

We now present new results using bidirectional Mendelian randomization analyses to address the possibility of reverse causation and false-positive findings.

Manuscript changes:

Lines 258-261: “Finally, we performed bidirectional or reverse MR on significant results to examine the potential presence of reverse causation. We selected instruments for each disease as described above from Pan-UKB and instruments for plasma TG levels from GLGC, essentially reversing the discovery stage design using a fixed-effect IVW method.”

Lines 368-373: “Finally, we performed reverse MR to estimate the effects of significant disease traits on plasma TG levels, selecting instruments from UKB and GLGC, respectively. Genetic data were sufficiently available to perform this analysis for 9 of the 19 diseases of interest. These results are presented in Supplementary File 6. Expectedly, “disorders of lipoprotein metabolism” and “hyperlipidemia” had positive effects on plasma TG levels; however, no other examined disease trait showed results suggesting reverse causation.”

2) The assumptions of Mendelian randomization need to be discussed in more detail.

We now state the assumptions of Mendelian randomization in the introduction section and discuss them in more detail in the Discussion section.

Manuscript changes:

Lines 124-129: “As in conventional MR, this method uses genetic variants as instrumental variables (IV) to proxy modifiable exposures (Davey Smithand & Ebrahim, 2003), and importantly, it relies on three critical assumptions: (1) The genetic variant is directly associated with the exposure; (2) The genetic variant is unrelated to confounders between the exposure and outcome; and (3) The genetic variant has no effect on the outcome other than through the exposure (Davey Smithand & Ebrahim, 2003).”

Lines 501-514: “The study has several limitations. First, MR is a powerful but potentially fallible method that relies on several key assumptions, namely that genetic instruments are (i) associated with the exposure (the relevance assumption); (ii) have no common cause with the outcome (the independence assumption); and (iii) have effects on the outcome solely through the exposure (the exclusion restriction assumption) (Hartwig et al., 2016). In MR, (i) is relatively straightforward to test, while (ii) and (iii) are difficult to establish unequivocally. As a prominent example, horizontal or type I pleiotropy has been shown to be common in genetic variation, which can bias MR estimates (Verbanck et al., 2018) (Jordan et al., 2019). This occurs when a genetic instrument is associated with multiple traits other than the outcome of interest. To detect and correct for this as best as possible, we used various MR tests as sensitivity analyses that each aim to adjust for or account for the presence of horizontal pleiotropy, including MR-PRESSO, as well as MR-Egger and weighted median methods. There is no universally accepted method that is perfectly robust to horizontal pleiotropy, but we take the best current approach by using multiple methods and examining the consistency of results.”

3) It is a strength to have a discovery stage and a replication stage. However, the difference between the two stages is blurred by the use of three tiers of statistical evidence. This may increase false positive rates. Further multiple testing adjustment may be needed.

We thank the reviewer for this important comment. Another Reviewer had also commented on this, and we provided a detailed response to clarify our approach. Please see Reviewer 3 comment below.

Manuscript changes: Please see Reviewer 3 comment below.

4) The interpretation of the causal effects may need to be more cautious and rigorous, particularly for the findings from Tiers 2 and 3.

We agree with the reviewers and have revised our language to moderate claims regarding causality, particularly for Tier 2 and 3 findings. Please see Reviewer 3 comment below.

Manuscript changes:

Lines 438-441: “Regarding non-ASCVDs, we present suggestive genetic evidence of potentially causal associations between plasma TG levels and uterine leiomyomas (uterine fibroids), diverticular disease of intestine, paroxysmal tachycardia, hemorrhage from respiratory passages (hemoptysis), and calculus of kidney and ureter (kidney stones).”

Lines 469-474: “However, we note that these associations were not as robust as those in Tier 1 associations related to ASCVDs, and we encourage caution in drawing causal conclusions from these estimates of a causal effect. Though we present rationale for potentially repurposing TG-lowering agents towards these non-ASCVD indications and for pursuing mechanistic studies interrogating TG biology, RCTs remain the gold standard for assessing causality and remain necessary for assessing clinical potential.”

Lines 565-567: “However, caution is still warranted in inferring causality, as MR depends on specific assumptions and the validity of those assumptions must be carefully assessed. Thus, diverse study designs remain necessary to triangulate evidence on the causal effects of plasma TG levels.”

Reviewer #3 (Recommendations for the authors):I have another concern or suggestion. One possible interpretation of the causal roles of triglycerides, the authors proposed, "the causal association we observe between plasma TG levels and ASCVD risk may be mediated by the concentration of apoB particles in TRLs, captured by plasma TG level measurements." The possible mediating role of apoB could be tested in the same multivariable MR framework used to exclude the effect of LDL-C and HDL-C. Since there are existing GWAS for apoB, it should be relatively easy to implement this analysis, either using the GWAS of apoB in UK Biobank and then outcome GWAS in FinGenn, or other GWAS of apoB (not in UKB) and then outcome GWAS in UKB.

We thank the reviewer for this suggestion to include ApoB in the multivariable MR analysis, given the atherogenic role of ApoB particles and their potential correlation with TG. We now include ApoB in the multivariable MR analysis and jointly estimate the independent causal effect of each exposure (TG, LDL, HDL, and ApoB plasma levels). Additionally, we present a cautious interpretation of these results in the Discussion section.

Manuscript changes:

We now overlay an additional forest plot (red) to Figure 4, which corresponds to the multivariable MR estimates after adjusting for LDL, HDL, and ApoB levels.

Accordingly, we have revised the caption for this figure: “Figure 4. Causal estimates of genetically proxied plasma TG levels on disease risk using multivariable IVW regression controlling for plasma LDL-C, HDL-C, and ApoB levels in UKB.”

Line 430-437: “We evaluated this possibility by accounting for ApoB levels in a multivariable MR framework and found that significant causal associations persist between plasma TG levels and all examined ASCVD traits, despite a modest decrease in effect size (Figure 4; Supplementary File 4). These data suggest that ApoB levels may explain at least a portion of the detected associations; however, plasma TG measurement appears to capture additional mechanisms of ASCVD risk involving cholesterol-enriched TRL remnants that remain unclear. Scientific interest in the atherogenic mechanisms of TRLs continues to grow, as recently reviewed elsewhere (Borén et al., 2022) (Ginsberg et al., 2021).”